# A 3D-Printed Soft Haptic Device with Built-in Force Sensing Delivering Bio-Mimicked Feedback

**DOI:** 10.3390/biomimetics8010127

**Published:** 2023-03-22

**Authors:** Rahim Mutlu, Dilpreet Singh, Charbel Tawk, Emre Sariyildiz

**Affiliations:** 1Faculty of Engineering and Information Sciences, University of Wollongong in Dubai, Dubai P.O. Box 20183, United Arab Emirates; 2Intelligent Robotics & Autonomous Systems Co (iR@SC), RA Engineering, Shellharbour, NSW 2529, Australia; 3Biofabrication and Tissue Morphology (BTM) Group, Centre for Biomedical Technologies, Faculty of Engineering, Queensland University of Technology, Brisbane, QLD 4000, Australia; 4School of Engineering, Department of Industrial and Mechanical Engineering, Lebanese American University, Byblos P.O. Box 36, Lebanon; 5Applied Mechatronics and Biomedical Engineering Research (AMBER) Group, School of Mechanical, Materials, Mechatronic and Biomedical Engineering, University of Wollongong, Wollongong, NSW 2522, Australia

**Keywords:** wearable robotics, soft haptics, soft sensing, 3D printing, biofeedback, bio-mimetics, human–computer interface, haptics, tactile feedback, proprioception

## Abstract

Haptics plays a significant role not only in the rehabilitation of neurological disorders, such as stroke, by substituting necessary cognitive information but also in human–computer interfaces (HCIs), which are now an integral part of the recently launched metaverse. This study proposes a unique, soft, monolithic haptic feedback device (SoHapS) that was directly manufactured using a low-cost and open-source fused deposition modeling (FDM) 3D printer by employing a combination of soft conductive and nonconductive thermoplastic polyurethane (TPU) materials (NinjaTek, USA). SoHapS consists of a soft bellow actuator and a soft resistive force sensor, which are optimized using finite element modeling (FEM). SoHapS was characterized both mechanically and electrically to assess its performance, and a dynamic model was developed to predict its force output with given pressure inputs. We demonstrated the efficacy of SoHapS in substituting biofeedback with tactile feedback, such as gripping force, and proprioceptive feedback, such as finger flexion–extension positions, in the context of teleoperation. With its intrinsic properties, SoHapS can be integrated into rehabilitation robots and robotic prostheses, as well as augmented, virtual, and mixed reality (AR/VR/MR) systems, to induce various types of bio-mimicked feedback.

## 1. Introduction

Haptics, as a combination of touch sensation (i.e., tactile) and kinesthetic feedback of the position of a limb, enables humans to interact with their surrounding environment [1]. Haptics has long been explored as an alternative communication method to substitute the lack of sensations such as touch and vision, which allow an individual to obtain the physical characteristics of any environment they interact with. Such haptic sensations can include the identification of the stiffness of an environment, texture, temperature, shape (i.e., form), location (i.e., position), weight, or a combination of those. The tactile sensing network covers the entire skin of a person with tactile nerve endings, which aids in discriminating between light and firm touch, and various surface textures of objects. Proprioceptors (i.e., proprioceptive receptors), on the other hand, are located in the joints, muscles, and tendons of the body. A haptic sensation is directly delivered to a user through a haptic interface (HI), which is a device that reproduces the required or desired sensation [2]. Besides the use of visual and auditory cues as common human–computer interfaces (HCIs), there is an increasing demand for other means of haptics, such as touch with a combination of force, motion, and vibrations, leading to creating artificial haptic sensations for more effective HCIs. Spatiotemporal trajectories [3], position trajectories [4], and contrary motions [5] can be effectively communicated to a user through the use of such haptic biofeedback, mediating kinesthetic and/or cutaneous sensations with mechanical stimuli. The perception of haptic stimuli is greatly influenced by the form of the percept, along with its location on the body, forces applied on the skin, its frequency, and the gender of the subject [6]. For instance, the frequency of the stimulus is very important in electrical stimulus-based haptics [7], while mechanical stimuli such as soft haptics can communicate similar percepts using lower frequencies by varying the amplitude of the stimulus [8].

Primary haptic explorations focus on methods and materials to induce biofeedback to create more effective HCIs for applications such as surgery robots [9]; rehabilitation robotic devices [10,11]; robotic prosthetics [12]; virtual, augmented, and mixed reality (VR/AR/MR) systems [13,14]; wearable robotics [15,16,17]; VR-haptics combined rehabilitation platforms [18]; and human movement augmentation [19]. The benefits of minimally invasive robotic surgeries include less trauma to the patient, reduced recovery time, and shorter post-surgery hospital stay [20]. Coupled with haptic feedback, also commonly known as proprioception, robot-assisted surgeries have enabled surgeons to perform teleoperations with improved perception compared with indirect vision-based operations [21]. VR/AR/MR technologies provide visual feedback using head-mounted display interfaces that are combined with auditory cues for the user (e.g., Oculus Rift, HTC VIVE, and GearVR) [22,23]. VR/AR/MR systems have been recently complemented with data gloves that map hand gestures from the user to the VR/AR/MR environment while providing tactile feedback from the VR/AR/MR environment back to the user [23,24,25]. Robot-assisted rehabilitation offers functional training with higher therapy intensity while providing controlled and assisted movements. Neurological disorders (e.g., stroke) reduce not only muscle strength but also proprioceptive sensations in the patient’s affected limbs [26]. Recovery, through a relearning process called neuroplasticity, is a phenomenon in which the brain tends to regain control over the movements of muscles by creating new neural paths in the brain [27]. Biofeedback systems play a significant role in improving neuroplasticity in the rehabilitation process of patients with neurological disorders by providing them with the necessary cognitive information that is not present due to the disorder [28]. Real-time biofeedback provided to the patients not only increases their motivation but also accelerates their neuroplasticity in the motor cortex, especially when biofeedback is synchronously delivered with therapeutic interventions [29]. Multisensory feedback, such as auditory, visual, and tactile feedback, accelerates motor learning and neuroplasticity, more effectively enhancing the recovery of patients [30]. These biofeedback systems can improve the sensing ability of patients relative to the kinematics, kinetics, and muscular activities of their affected limbs in the form of mechanical [31], electrical [32], and vibrational stimulation [33] with the use of different actuation and sensing technologies, and materials such as piezoelectric materials [34], polymers [35], elastomers [36], pneumatics [37], and/or a combination of them [38]. Vibrotactile stimulation is frequently utilized; it employs vibration motors, which can be worn onto a predefined location on the patient’s body [39]. While vibrotactile stimulation systems are identified as potential biofeedback methods, as they are noninvasive and low-cost systems, they need to be combined with other mechanical components, such as eccentric rotating mass, to improve stimulation density. In addition, vibrotactile stimulation systems need to be equipped with low-elastic-modulus materials for comfort, and they require sophisticated control algorithms. Soft haptics realized with soft robotics principles and concepts in terms of actuation, sensing, modeling, and materials, on the other hand, is yet to be studied for biofeedback systems.

Soft robotic systems are advantageous with respect to traditional robotic systems due to their adaptability, conformability, agility, and high compliance [40,41]. More prominently, they can safely interact with humans [42], operate in highly dynamic environments, and uniformly conform to unstructured environments and/or objects with very low contact forces [43,44]. Using these intrinsic properties, soft haptic devices with actuation and sensing capabilities integrated into a single monolithic, soft structure can be developed. Actuation concepts for soft robots can be classified into two major groups: actuation due to material properties (i.e., smart materials) and external actuation (e.g., soft pneumatics). Soft pneumatic and hydraulic actuators are common soft robotic actuation concepts, in addition to tendon-driven soft structures. Pneumatically actuated soft robots have been extensively developed due to their large-range and multimodal deformation, and their ability to mimic their natural counterparts. A soft haptic armband based on soft pneumatic actuators as a feedback system can enhance the user perception of a teleoperated robotic hand delivering information from the environment back to the user in a safe manner [45]. Similarly, a wearable haptic HCI employs pneumatic soft actuators to deliver sensory information to the wearer [46]. A compact, easy-to-fabricate, and easy-to-implement haptic device is yet to be developed using soft robotic actuators and sensors.

The majority of flexible and/or soft sensors are either resistive or capacitive. Soft capacitive sensors are usually composed of an elastomer as the dielectric material that is sandwiched between two conductive electrodes [47]. Soft resistive sensors, on the other hand, are usually made of an elastomeric matrix that is filled with a conductive material [48] or microchannels within an elastomeric structure that are filled with conductive inks [49]. Force sensing resistors (FSRs), flexible force sensors that are usually used in the limb–socket interface of robotic prostheses, are commercially available [21]. However, these flexible and/or soft force sensors require complicated fabrication methods and have yet to be combined with a more compact soft haptic device.

The main contributions of this study are (i) the design and development of a unique soft haptic system, SoHapS, which combines pneumatic soft actuation with soft force sensing capabilities to deliver a haptic interface with a safe HCI; (ii) combined mechanical and electrical characterization; (iii) a dynamic model of the soft haptic device to estimate its force output; and (iv) validation of the static and dynamic characteristics with biofeedback applications for tactile and proprioceptive feedback using a soft mechano-tactor in the form of mechanical pressure rather than vibration. In this study, we focus on developing a soft haptic system and demonstrate its capabilities to substitute tactile signals and proprioception, which are critical to developing intuitively controllable robotic applications with a soft HCI. Compared with existing non-invasive haptic systems, SoHapS embraces simplicity in design and pneumatic actuation; is fully 3D-printed; requires no sophisticated mechanisms of assembly to communicate the force/torque from an actuator to the wearer’s skin; has a high range of force/pressure; can adjust to the sensitivity of the wearer’s perception; and can measure the direct force applied to the wearer with its built-in, custom, 3D-printed soft force sensor. Our soft haptic system integrated with a custom soft resistive force sensor was fully 3D-printed using a fused deposition modeling (FDM) 3D printer. In one of our previous pilot studies, we introduced the concept of using a soft haptic device, in which multiple soft haptic units were combined with external force sensitive resistors (FSRs), to improve postural balance in subjects [50]. In this study, we demonstrate the efficacy of SoHapS with not only tactile but also kinesthetic sensory substitution with the use of the identified static and dynamic characteristics. The major focus is centered on the extensive characterization of the soft haptic biofeedback device; we characterized it both mechanically and electrically to assess its performance, and a dynamic model was developed to predict its force output for given pressure inputs. The characterized soft haptic biofeedback device was then validated with demonstrations of its capacity to mimic biofeedback in the form of tactile feedback, such as gripping force, and proprioceptive feedback, such as finger flexion–extension positions, in the context of teleoperation. With the use of dual-nozzle printing, the soft resistive force sensor was fabricated with soft and conductive thermoplastic polyurethane (TPU) with carbon black filler, while the soft actuator (i.e., bellow-shaped soft pneumatic actuator (SPA)) was fabricated with soft and flexible thermoplastic polyurethane (TPU). The customized design of SoHapS was optimized for fabricating airtight actuators with the desired performance, as depicted in Figure 1. Static and dynamic properties were characterized, and showed linearity of up to 150 kPa pressure input, low hysteresis, and great repeatability.

## 2. Materials and Methods

A single SoHapS unit consists of three major components: (1) an actuation component consisting of a soft, positive-pressure, bellow actuator; (2) a structural component consisting of a soft case with an interchangeable tip; (3) a sensing component consisting of a soft force sensor, as shown in Figure 1. The soft sensor was 3D-printed using commercially available, flexible, conductive TPU (PALMIGA PI-ETPU 95-250 Carbon Black; Creative Tools, Sweden), and the bellow actuator and the soft case were 3D-printed using another commercially available soft TPU type known as NinjaFlex (NinjaTek, Lititz, PA, USA).

### 2.1. Design, Fabrication, and Instrumentation of SoHapS

The computer-aided design (CAD) models were created and modeled using Creo Parametric 2.0 (PTC Inc., Boston, MA, USA) and were sliced using a commercially available slicer (Simplify3D; Simplify3D LLC., Blue Ash, OH, USA). The parts were 3D-printed using an open-source FDM 3D printer (FlashForge Inventor, FlashForge Corporation, Industry, CA, USA) using 100% infill. The printing parameters optimized for obtaining airtight and homogeneous soft structures are provided in Appendix B. A 6-axis force sensor (K6D27; ME-Meßsysteme GmbH) was used to obtain the blocked force response to various pressure inputs. A positive-pressure pneumatic pump (AEG Mini Wheelbarrow Compressor; 19 L; 10 Bar) was employed to actuate SoHapS. The real-time pressure regulation and recording of the sensory output were carried out directly with Simulink (The MathWorks Inc., Natick, MA, USA) using a data acquisition board (HUMUSOFT MF644, Humusoft Ltd., Praha, Czech Republic). The tip force output of SoHapS was simultaneously measured with the output of the soft sensor.

### 2.2. Finite Element Analysis (FEA)

Finite element modeling (FEM) simulations were performed on a single SoHapS device to predict its performance in terms of blocked force, deformation, strain, and equivalent stress. ANSYS Mechanical (ANSYS Inc., Canonsburg, PA, USA) was used to perform the FEM simulations. A 5-parameter Mooney–Rivlin hyperelastic material model and a 3-parameter Mooney–Rivlin hyperelastic material model were used for NinjaFlex and conductive TPU, respectively. The material models were obtained based on the experimental stress–strain data of NinjaFlex and conductive TPU [42]. The material model parameters for Ninjaflex were C10 = −0.23 MPa; C01 = 2.56 Mpa; C20 = 0.12 Mpa; C11 = −0.56 Mpa; C02 = 0.9 Mpa; and D1 = 0 Mpa^−1^. For conductive TPU, they were C10 = 0.58 Mpa; C01 = 4.37 Mpa; C11 = −0.047 Mpa; and D1 = 0 Mpa^−1^. The SoHapS assembly was meshed using higher-order tetrahedral elements. Fixed support boundary conditions were imposed on the base and at the tip of the SoHapS unit. A set of pressure inputs was imposed normally to the internal walls of the bellow actuator, as shown in Figure 2a. Contact pairs were defined between the adjacent walls of the assembly that were initially in contact and the ones that came into contact upon deformation. A 0.6 friction coefficient, which is sufficient for TPU-based materials, was applied for contact pairs. Pure penalty formulation was identified for the contact model, with no slipping allowance.

The behavior of SoHapS was simulated using FEM at an applied pressure of 150 kPa. Figure 2b shows the initial model before any applied pressure, and Figure 2c,d show the FEM deformation of SoHapS in its deformed state after applying the pressure of 150 kPa. The deformation behavior was used to design SoHapS in a way to ensure that it underwent homogenous deformations in its overall structure.

FEM was also utilized to analyze the contact behavior of the soft force sensor, as shown in Figure 2e–h. With higher pressure inputs, contact separation was observed in the soft force sensor. The contact was initially established as expected, in the middle of both parts, as presented in Figure 2f. However, larger deformation occurring in the head part of the sensor resulted in a separation of contact in the middle of the contact region under larger input pressures. The change in the contact area affects the change in the overall resistance of the sensor. Thus, the final design was optimized to ensure that the applied pressure was within the limits of contact separation. Upon applying input pressure to SoHapS, it generates a compression force deforming the sensor parts, thus changing the resistance of the soft force sensor (Figure 2f–h).

### 2.3. Characterization and Modelling

The static characteristics of SoHapS were studied in terms of blocked force, linearity, sensitivity, hysteresis, and signal-to-noise ratio; the dynamic characteristics were evaluated under various loading conditions by fitting a linear transfer function to the input and soft force sensor output. The change in the soft force sensor response (i.e., output signal) occurs due to deformation in the sensor, which is due to the force generated by the SPA of SoHapS when a pressure input is applied. We tested the soft sensor response with a stair-stepped pressure input to assess its linearity and thus validated the use of the linear system identification method. The proposed 3D-printed soft force sensor showed a linear response. We obtained the dynamic response of SoHapS under various loading conditions, as demonstrated in Section 3. The transfer function is the ratio of the Laplace-transformed system output to the Laplace-transformed system input, which is generally in step, impulse, ramp, and sinusoidal forms, with the assumption that the initial conditions are zero, i.e., P_0_ = 0 kPa and R_0_ = 0 Ω. Thus, the transfer function of SoHapS to be identified is as follows:(1a)$$G\left(s\right)=\frac{L\left[sensor\;response\right]}{L\left[pressure\right]}=\frac{R\left(s\right)}{P\left(s\right)}$$
(1b)$$\frac{R\left(s\right)}{P\left(s\right)}=\frac{b_{0}s^{m}+b_{1}s^{m-1}+\cdots +b_{m-1}s+b_{m}}{a_{0}s^{n}+a_{1}s^{n-1}+\cdots +a_{n-1}s+a_{n}}$$

This approach not only helps to determine an empirical transfer function for SoHapS but also provides the ability to estimate dynamic characteristics such as the bandwidth of the system. We also studied the relation between the force output and the soft sensor response, which can be used to determine the force exerted directly onto the skin of the wearer by SoHapS. The identification of the dynamic characteristics was carried out using MATLAB System Identification Toolbox (R2020b). We applied positive pressure to SoHapS and simultaneously measured its voltage output of the soft force sensor and the force output using the 6-axis force sensor, as shown in Figure 3.

## 3. Experimental Results and Discussion

We conducted two series of experimental tests: (i) a series of experimental tests conducted to characterize SoHapS and (ii) two different bio-mimicked-feedback applications demonstrating the efficacy of SoHapS, as demonstrated in Section 4. The pressure input was dictated using a Simulink model and generated using a proportional pressure regulator (Festo VPPM-6L-L-1-G18-0L10H-V1P-S1C1), which was attached to SoHapS. The six-axis force sensor was also positioned at the tip of SoHapS in direct contact to measure the force exerted at the tip. SoHapS was subjected to various pressure input values ranging from 0 to 250 kPa, with a step of 50 kPa, in the form of steps/stairs and sinusoids at the different frequencies of 0.027, 0.053, and 0.132 Hz. The performance of SoHapS was analyzed under steady-state and dynamic conditions to study its response as described in Section 2.3. The airtightness of the 3D-printed SPA was verified prior to any experimental tests. The 3D-printed SPAs were inflated whilst submerged in water to verify that they were completely airtight. The experimental setup for the characterization process is shown in Figure 3. Each input was applied for 30 s so that the soft sensor reached its steady-state response and the transient response of SoHapS could be recorded. Along with various advantages of using pneumatic actuators, they bring some limitations due to the components required to generate the necessary air pressure. These limitations include an air compressor, proportional pressure regulators, and pneumatic hoses, which bring bulkiness to the overall experimental setup, in addition to airtightness, which plays a key role in obtaining functional actuators, especially in 3D-printed soft robotic actuators. Nevertheless, most of these limitations are addressed in many ways. For instance, a bulky air compressor can be replaced with a mini air compressor with pressure ability under 150 kPa, a slimmer pneumatic hose, and miniature pressure regulators.

### 3.1. Steady-State Characteristics

#### 3.1.1. Blocked Force

The blocked force of the SoHapS unit was measured using a six-axis force sensor. SoHapS was subjected to variable pressure values of 0–250 kPa with 50 kPa increments, and the blocked forces were measured along with the corresponding electrical resistance change of the soft force sensor, as shown in Figure 4. The compressive blocked forces are presented as positive values in Figure 4.

#### 3.1.2. Linearity and Sensitivity

The soft force sensor parts were 3D-printed in a way to allow the current to flow in a direction parallel to the printing direction. Only axial deformations were allowed for the actuator to minimize and prevent any contributions to the resistance change due to lateral deformations. The electrical response of the sensor depends on the contact area between the conducting parts. When SoHapS was pressurized, the contact area between the conductive parts increased, which resulted in a decrease in the electrical resistance of the sensor. The design of the conducting parts and the 3D-printing parameters influence their corresponding electrical resistance, as outlined in Appendix B. The soft force sensor is instrumented with a voltage divider to obtain its resistance values, which are then correlated to the pressure applied and the force generated at the tip of SoHapS. The soft sensor is connected to the circuit with the sensor base and sensor head. As exhibited in Figure 2e,f, the contact area affects the resistance values of the soft force sensor.

The steady-state values of the blocked force and the change in the soft sensor response were used to obtain the steady-state relationships between the input pressure and its corresponding blocked force, and between the same input pressure and its corresponding electrical resistance change of the soft force sensor, as shown in Figure 5. Linear fits were used to determine the linearity and the sensitivity of the force and electrical resistance responses, as shown in Figure 5. The steady-state response of the soft sensor of SoHapS showed a linear response up to 150 kPa, regardless of the speed of pressure stepping. Sensitivity indicates the minimal change in the measured variable (i.e., the resistance changes in the soft force sensor) due to the pressure input, which is defined as the ratio of the change in the output to the change in the input.
(2)$$S=\frac{\Delta \left(\Delta R/R_{0}\right)}{\Delta p}$$

The sensitivity of the soft force sensor was calculated to be 0.3268 kPa^−1^.

#### 3.1.3. Hysteresis

The investigation of the effect of hysteresis was conducted on SoHapS during the loading and unloading phases, as shown in Figure 6. The results obtained show that the hysteresis effect increased from ≈7% to ≈30% as the operating frequency increased from 0.027 Hz to 0.053 Hz. The viscoelastic properties of the material used and the stiffness of the soft sensor of the SoHapS design greatly influence the hysteresis effect.

#### 3.1.4. Signal-to-Noise Ratio

The base resistance of the circuit was recorded as no-load resistance and used as reference resistance (R_o_) for calculations. The average resistance of the SoHapS unit in the initial position when no pressure input was applied was 0.784 MΩ, as shown in Figure 7. Some fluctuations in electrical resistance were observed, which is typical for resistive materials. The signal-to-noise ratio for the soft force sensor of SoHapS under no-load conditions was calculated to be 26.892 dB.

### 3.2. Dynamic Response Characteristics

SoHapS was subjected to several pressure signals at various amplitudes and frequencies during dynamic response characterization.

#### 3.2.1. Cyclic Input Response

The force and electrical resistance responses of SoHapS were studied with sinusoidal pressure inputs at various peak-to-peak amplitudes and frequencies, as shown in Figure 8.

#### 3.2.2. Dynamic Model Estimation

The dynamic model estimation of SoHapS was performed using the electrical response of the soft force sensor, where first- and second-order systems were employed to predict the response of SoHapS with a match of over 90%, as shown in Figure 9. The dynamic models of the resulting transfer functions are summarized in Equations (3) and (4). The first-order transfer function,
(3)$$G_{1}\left(s\right)=\frac{0.6218}{s+2.49}$$
estimates the experimental data of the soft sensor response with a 91.46% match, while the second-order transfer function,
(4)$$G_{2}\left(s\right)=\frac{1.463\;s+0.1266}{s^{2}+6.352\;s+0.4951}$$
estimates the experimental data with a 94.04% match. Bode plots are shown in Figure 10 for both the first-order and second-order transfer functions.

#### 3.2.3. Validation

Dynamic model estimation was conducted with a sinusoidal pressure input at 170 kPa peak-to-peak amplitude and 0.027 Hz frequency. Model validation with the transfer functions was carried out using the soft sensor electrical resistance response with a signal at different amplitudes and 0.053 Hz operating frequency. $G_{1}\left(s\right)$ was validated with 88.51% fitting, whereas $G_{2}\left(s\right)$ estimated 90.6% similarity, as shown in Figure 11.

## 4. Inducing Bio-Mimicked Feedback with SoHapS

We developed two different demonstration tasks for validating the static and dynamic characteristics with biofeedback applications for tactile and proprioceptive feedback using a soft mechano-tactor in the form of mechanical pressure and for highlighting the efficacy of SoHapS in inducing substitute bio-mimicked feedback. This study focused on mimicking tactile feedback with alternative means for delivering external feedback to the subject rather than attempting to replicate tactile feedback at higher frequencies. SoHapS substitutes biofeedback with the use of mechanical pressure intensity in contrast to the vibrotactile approach. The first task mimicked and communicated the gripping force that the primary subject (i.e., S1) applied at the fingertip. This fingertip force was rendered as tactile feedback to the second subject (i.e., S2) using three discrete states of gripping force. The second task replicated the proprioceptive feedback of the index-finger flexion–extension of S1 using a flex sensor. The proprioceptive feedback was rendered as a mapping of the finger bending angles (i.e., primarily metacarpal joint (MCP), since a single sensor and a single SoHapS device were used). S1 randomly flexed (bent) their finger while measuring their angular finger position. The measured finger (i.e., MCP joint) positions were simultaneously communicated to S2 with mapped pressure inputs using SoHapS. Both experiments were conducted in a setting where S1, shown on the right in Figure 12a, performed the tasks as the master subject, and S2, shown on the left in Figure 12a, was stimulated by the input of SoHapS on the wrist (see Figure 12a). S2 predicted and pointed to the level of gripping force in the first task for tactile feedback and flexed–extended their index finger in accordance with the proprioceptive feedback received. The vision of S2, wearing SoHapS, was blocked with an opaque piece of paper in the middle of the table to prevent any visual feedback of S2 acting as the master. In addition, S2 was unable to hear the other subject performing actual tests (i.e., gripping and finger movements), as S2 wore noise-canceling headphones.

In both experiments, the participants sat apart, and a large opaque piece of paper was set up in the middle of the table to block the vision of S2. In addition, S2 wore noise-canceling earphones to eliminate any influence created by experimental equipment, particularly the pressure regulator. In the tactile feedback task, a 3D-printed cube was equipped with a force resistive sensor (FSR) to measure the gripping force (Figure 12c). The 3D-printed cube was designed with a pocket to accommodate a weight. In the proprioceptive feedback task, the flex sensor was fitted to the index finger of S1 (Figure 12d). Both FSR and flex sensors were instrumented with a voltage divider to accommodate the measured gripping force and index finger position and to map these sensory measurements as SoHapS-bio-mimicked feedback to S2. The minimum detectable and maximum comfortable input pressures were determined before these experiments and mapping. A digital camera was set up to record the tests, which were then analyzed to determine the efficacy of SoHapS in inducing two distinct bio-mimicked feedback types (Appendix A).

### 4.1. Communicating Gripping Force as Tactile Feedback

In this task, SoHapS was strapped tightly to S2’s left wrist, while S1 gripped the cube, which was equipped with an FSR force sensor. The choice of the hand on which SoHapS was worn was left to the subject with no predetermination. With respect to the state of the cube, it was either empty, loaded with a weight of 100 g, or not being gripped (Figure 12c), and the gripping force applied by S1 changed. This gripping force obtained by the FSR sensor was communicated to S2 and mapped with SoHapS activation.

Before the actual tactile feedback experiment, S2 was trained three times for each state, and states were verbally communicated. Then, in the actual test, S2 wore noise-canceling earphones. While S1 performed the task of gripping the cube, S2 pointed to pre-printed illustrations; no load (gripping), empty cube, and loaded cube. The analyzed frames from recorded videos showed that SoHapS could be used to induce substitute gripping force (Appendix A). The reaction times of S2 and the accuracy of the interpretation were obtained and are outlined in Figure 13a, where *w* depicts weighted-cube gripping, *e* depicts empty-cube gripping, and *o* depicts no gripping, with the markers of full square, empty square, and crossed square, respectively, being used to indicate the corresponding timings. While the reaction times varied, with an average of 0.647 s, S2 only misinterpreted once, which was the 12th test point, interpreting empty-cube gripping as weighted-cube gripping.

### 4.2. Communicating Proprioceptive Feedback

As in the previous test, SoHapS was strapped tightly to S2’s left wrist, while S1 wore the flex sensor on their index finger. The choice of the hand on which SoHapS was worn was left to the subject with no predetermination. Both participants placed their index fingers on printed protractors. S1, wearing the flex sensor, placed their index finger on a reference protractor and flexed the finger hovering on the protractor. The angular position of the finger flexion of S1 was measured by the flex sensor. S2 wore SoHapS on the left hand, (i.e., hand choice was arbitrary) and placed the index finger of their other hand on another protractor at their side (as shown in Figure 12a). Prior to the actual tactile feedback experiment, S2 was trained three times for each state of four finger flexion positions (from fully relaxed to fully flexed), and the states were verbally communicated. Then, in the actual test, S2 wore noise-canceling earphones. While S1 randomly flexed and extended their index finger in the four flexing states, S2 predicted the finger position of S1 by mimic-flexing their index finger of the other hand. The analyzed frames from recorded videos showed that SoHapS could be effectively used to induce substitute proprioceptive feedback (Appendix A).

The results are summarized in Figure 13b as reaction times, finger flexion angle of both subjects, and interpretation error of S2. Comparing this test to the previous test, this test required not only several finger positioning states but also randomly selected finger angular positions within a range of −10–90°, and SoHapS-bio-mimicked-feedback pressure was mapped to the range of finger angular positions, in which −10° was communicated as no pressurization. S2 showed great interpretation of finger flexion angles, with interpretation error of less than 10° and consistent average reaction times of 0.797 s with minimal pre-training. Both experiments for inducing different bio-mimicked-feedback types were conducted to demonstrate the efficacy of SoHapS. We noticed that there was a delay in the predictions of S2 of gripping force and index finger positions of S1, which should be thoroughly studied before implementing SoHapS in control systems, such as rehabilitation robots, robotic prostheses, or VR/AR/MR systems, in consideration of system delays.

## 5. Conclusions and Outlook

Although haptics has been extensively studied, there is significant interest, as well as gaps, in inducing bio-mimicked feedback to create more effective communication between humans and robots/computers. Wearable soft haptic systems hold great potential due to their intrinsic properties. We (i) designed and developed a new soft haptic system, SoHapS, which combines pneumatic soft actuation with custom soft force sensing capabilities to deliver a haptic interface with a safe HCI; (ii) combined mechanical and electrical characterization; (iii) obtained a dynamic model of the soft haptic device to estimate its force output; and (iv) validated the static and dynamic characteristics with biofeedback applications for tactile and proprioceptive feedback using a soft mechano-tactor. A low-cost and open-source FDM 3D printer was employed with the use of dual-nozzle printing, and a soft force sensor was fabricated with soft and conductive thermoplastic poly(urethane) (TPU) with carbon black filler, while the soft bellow SPA was fabricated with soft and flexible thermoplastic poly(urethane) (TPU). Both actuator and sensor parts were simulated using finite element modeling (FEM) to ensure the performance of SoHapS. Monolithic fabrication was adapted to simplify the fabrication procedure while achieving an integrated robotic system. Static and dynamic characteristics were demonstrated with two pilot tests in the context of teleoperation applications to deliver two different haptic feedback types to the wearer: (i) tactile feedback and (ii) proprioceptive feedback. In the tactile feedback application, the gripping force of Subject 1 was communicated to Subject 2, who wore SoHapS. In the proprioceptive feedback application, the index finger position of Subject 1 was communicated to Subject 2, who wore SoHapS and received the discrete soft forces applied from SoHapS.

In future work, we will expand our experimental studies on SoHapS to further optimize its performance and utilize it, in particular, in wearable rehabilitation robotic systems to study its efficacy and to improve the recovery and rehabilitation experience of patients with neurological disorders, such as stroke. Further characterizations are also required to estimate a dynamic model that can be robust at different operating frequencies. Future explorations will also include the use of multiple SoHapS devices to investigate the device influence on improving the actions of the wearer.

## Figures and Tables

**Figure 1 biomimetics-08-00127-f001:**
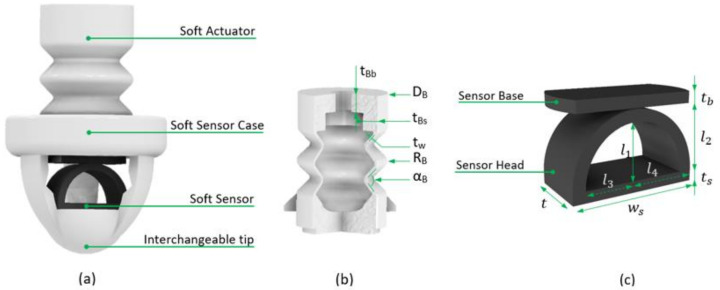
Three-dimensionally printed soft haptic system (SoHapS): (**a**) SoHapS complete system. (**b**) The actuation component of SoHapS consists of a bellow actuator with the following dimensions in mm: D_B_, 14.00; t_Bs_, 2.87; t_w_, 0.45; RB, 1.12; α_B_, 90°. (**c**) The sensing component of SoHapS consists of a soft force sensor of semi-elliptical shape with the following dimensions in mm: t_s_, 1.00; t_b_, 1.00; t, 6.00; w_s_, 11.40; l_1_, 4.95; l_2_, 6.00; l_3_, 4.60; l_4_, 5.70.

**Figure 2 biomimetics-08-00127-f002:**
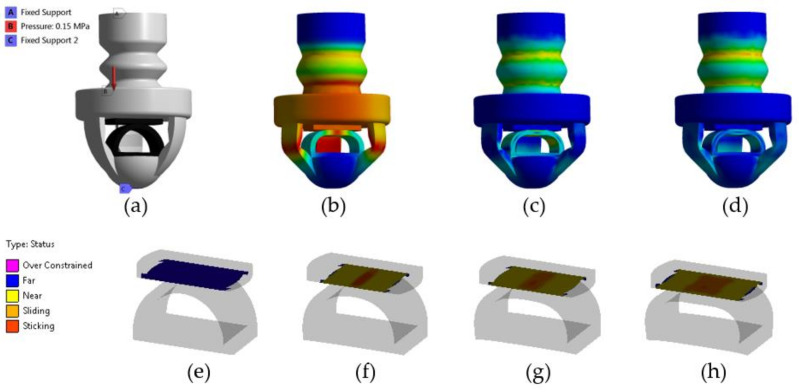
FEM results of SoHapS: (**a**) boundary conditions (A and C were fixed) at pressure of 150 kPa (B) applied to the internal walls; (**b**) deformation (max deformation in red, 1.583 mm); (**c**) equivalent stress (max stress in red, 3.628 MPa); and (**d**) equivalent strain (max strain in red, 26%). Changes in contact status of soft force sensor at (**e**) 0 kPa, (**f**) 50 kPa, (**g**) 100 kPa, and (**h**) 150 kPa pressure applied to SoHapS.

**Figure 3 biomimetics-08-00127-f003:**
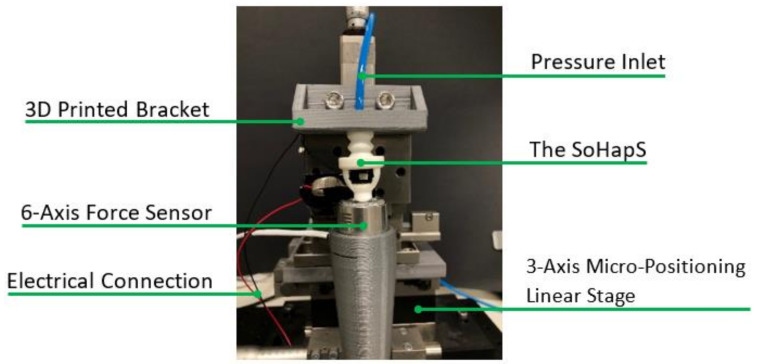
Experimental setup for characterization of SoHapS.

**Figure 4 biomimetics-08-00127-f004:**
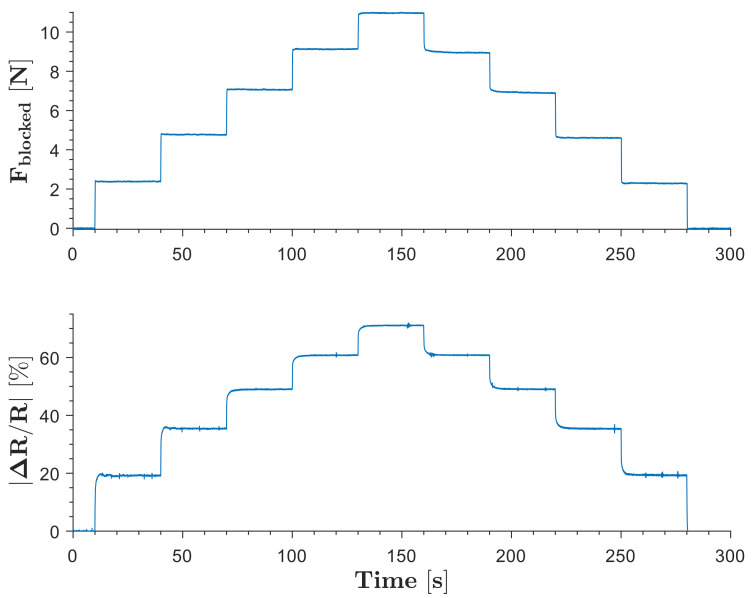
Blocked forces of SoHapS under various pressure inputs ranging from 0 to 250 kPa with 50 kPa steps. The primary axis shows the blocked force, and the secondary axis shows the change in resistance as the soft sensor response.

**Figure 5 biomimetics-08-00127-f005:**
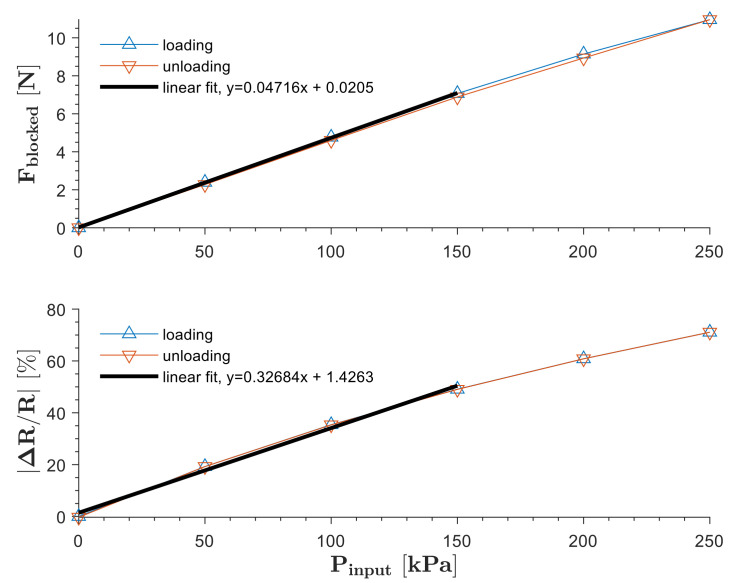
Linearity of SoHapS for blocked force and soft sensor response under stepping pressure inputs; linear fits for blocked force and sensor response (R^2^ = 0.9999 and R^2^ = 0.9942, respectively) are provided for up to 150 kPa due to the nonlinear response of the soft sensor above the pressure input of 150 kPa.

**Figure 6 biomimetics-08-00127-f006:**
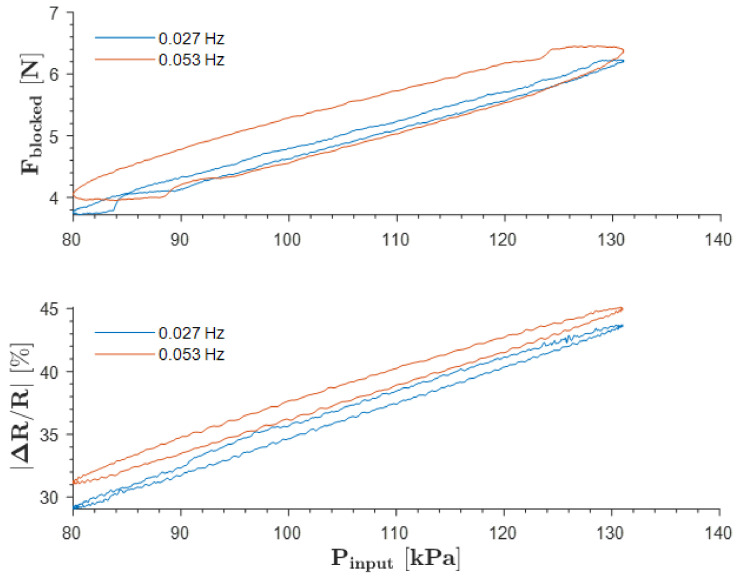
Hysteresis of SoHapS under peak-to-peak 50 kPa input pressure at operating frequencies of 0.027 Hz and 0.053 Hz.

**Figure 7 biomimetics-08-00127-f007:**
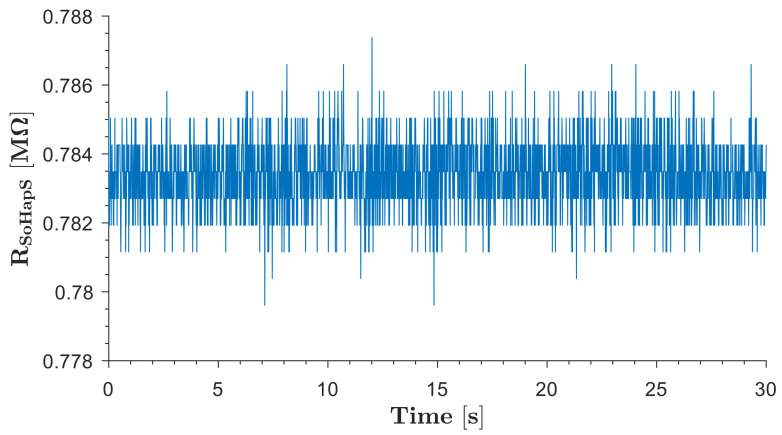
Electrical resistance of soft force sensor of SoHapS when no load was applied and at a 100 Hz sampling rate.

**Figure 8 biomimetics-08-00127-f008:**
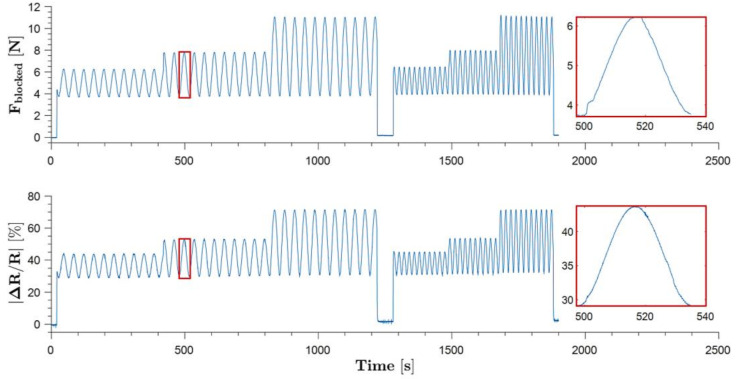
Behavior of SoHapS under cyclic loading at different amplitudes and frequencies (inset plots are provided for the period of a 0.027 Hz cycle).

**Figure 9 biomimetics-08-00127-f009:**
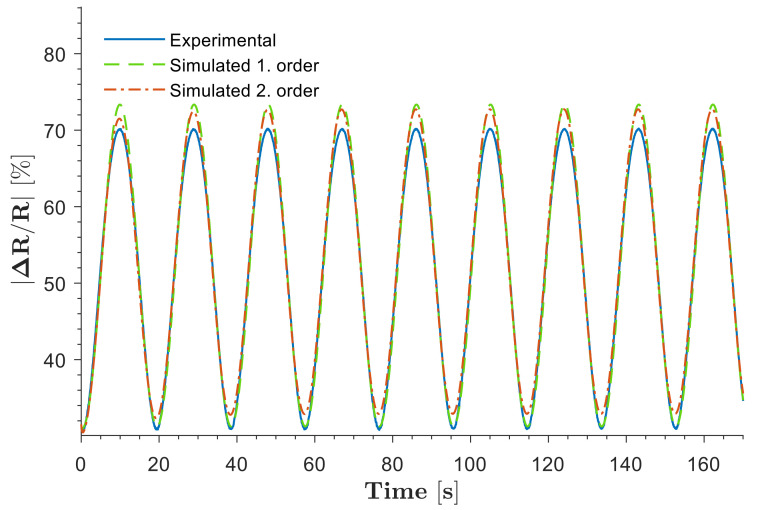
Comparison between the actual experimental response of SoHapS and the dynamic models.

**Figure 10 biomimetics-08-00127-f010:**
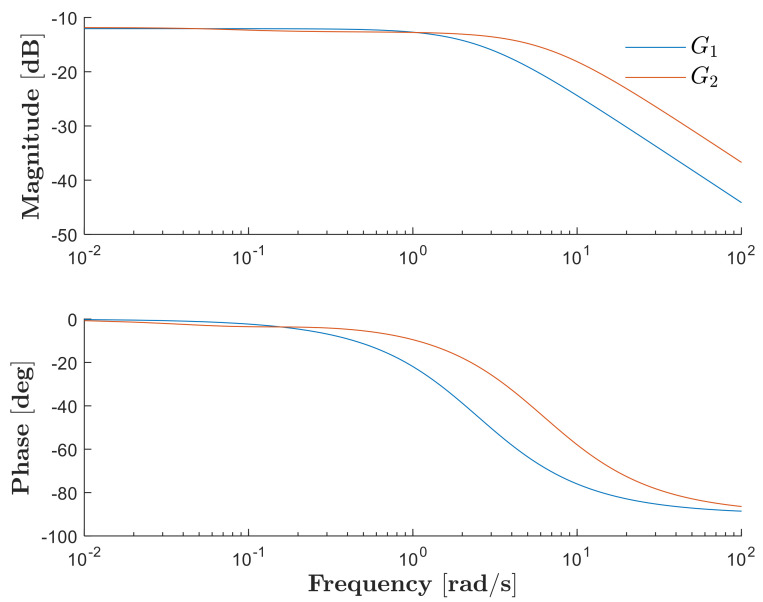
Bode plots of $G_{1}\left(s\right)$ and $G_{2}\left(s\right)$.

**Figure 11 biomimetics-08-00127-f011:**
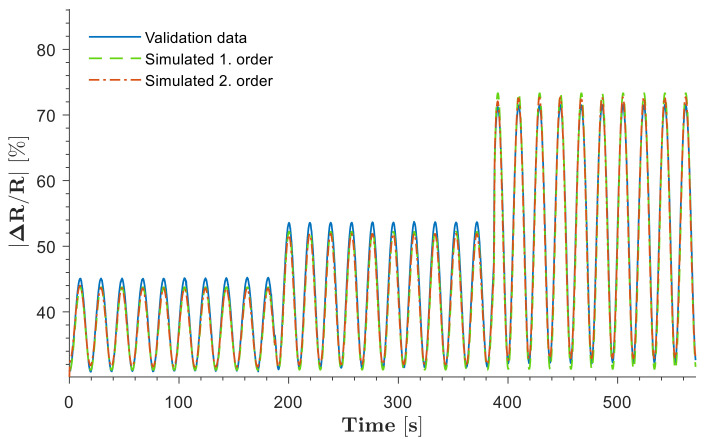
Dynamic model validation under various cyclic loading conditions.

**Figure 12 biomimetics-08-00127-f012:**
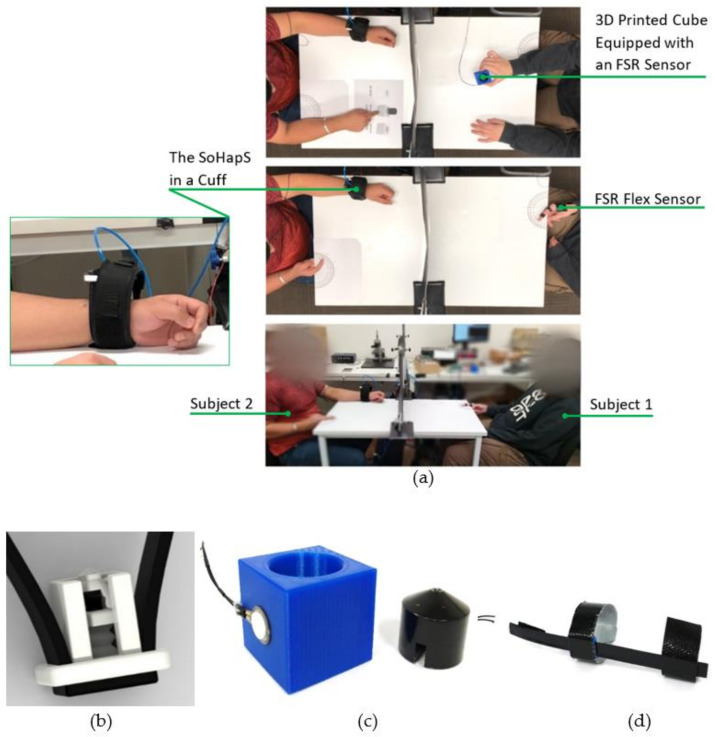
Experimental setup for (**a**) testing SoHapS in delivering substitute tactile feedback to communicate gripping force and proprioceptive feedback to communicate finger position (i.e., flexion–extension). (**b**) Three-dimensional CAD model of SoHapS with custom-designed housing and soft strap, (**c**) 3D-printed gripping force cube equipped with an FSR sensor, and (**d**) a flex sensor worn on the index finger by S1.

**Figure 13 biomimetics-08-00127-f013:**
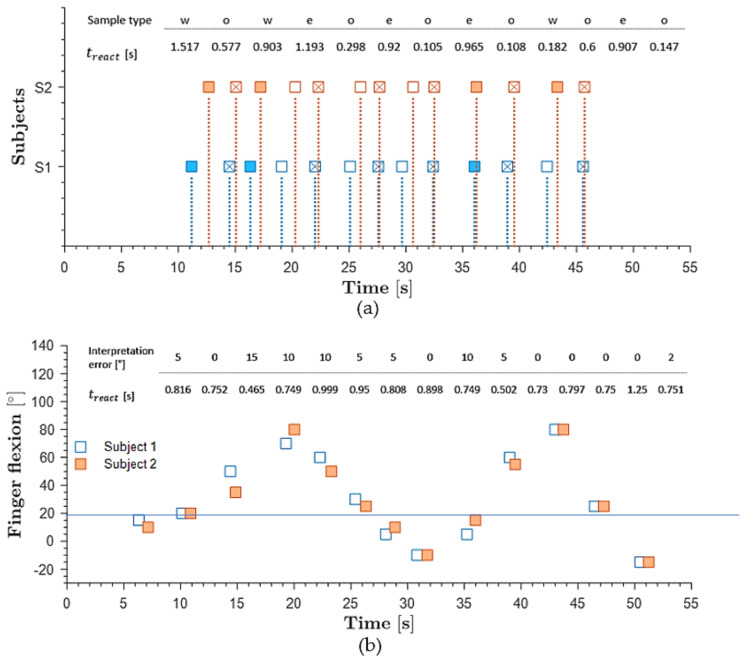
(**a**) Tactile feedback, with Subject 1 (S1) representing the master and Subject 2 (S2) playing the slave role in predicting the degree of index flexion of S1. ***w*** depicts weighted cube gripping; e depicts empty-cube gripping; and ο depicts no gripping. (**b**) Proprioceptive feedback with finger flexion angle interpretation error of S2 including reaction times.

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
