# Peer review of "A 3D-Printed Soft Haptic Device with Built-in Force Sensing Delivering Bio-Mimicked Feedback"

_biomimetics, 2023, doi:10.3390/biomimetics8010127_

Round 1

Reviewer 1 Report

The paper presents the design and characterization of an integrated soft haptic actuator and sensor. The design is 3D printable and is actuated pneumatically. Modeling and characterization demonstrate the ability of the proposed design to provide perceivable force feedback (tactile and proprioceptive). While the design is clearly presented and the characterization is sufficient (force feedback, sensitivity, linearity, SNR, hysteresis, etc.), there are several concerns that must be addressed before publishing this study. Here is a summary of my comments (I lean more towards rejecting the paper than accepting it due to the novelty issue):

Major comments:
1. The novelty of the design seems rather incremental. For example, I wonder what the fundamental differences between the proposed design and the one presented in the following study: https://www.mdpi.com/1424-8220/22/10/3779. It seems to have a very similar design (including an overlap of the characterization). This previous study is not even cited in the manuscript!

2. Putting the previous study aside, I also wonder what the fundamental contributions to the field of haptics/soft robotics are. While having a 3D printed design definitely brings convenience/saving in the fabrication process, however in terms of actuator/sensing functions, what is novel in the proposed design is not entirely clear to me.  

3. Section 4 evaluates the SoHapS device with human subjects, there are several questions/concerns. First of all, how many participants were recruited for the study (only 2?)? If only two subjects were recruited, then Figure 13 would not have statistical significance/ What are the demographics of the test subjects (gender, age group, hand orientation, etc.)? Was the study approved by an appropriate IRB committee? These questions must be addressed before properly analyzing the presented results.

4. The Authors claim that the design is capable of rendering tactile and force feedback. In order to render tactile feedback, vibrotactile stimulation should also be possible. To my understanding, that’s not true for the presented design (would it be possible to generate vibration feedback at a frequency of 300 Hz!?).

5. The Authors claim that the device provides a “”high range of force/pressure”feedback. However, the results demonstrate linearity up to around 150 kPa (equivalent to less than 8N). Is this really high force for a kinesthetic haptic device?

6. One fundamental requirement in the design of force feedback devices is its ability to generate multiple DoF force feedback. The current design provides a single DoF (to my understanding). Any thoughts on the potential of the proposed approach to extend to multiple DoF, including torque feedback?
7. It is not clear how the electric circuit is connected to the SoHapS device (in order to actively actuate it or to read the force measurement). Fig. 12 shows the device but does not show the connection to the control circuitry.

Minor comments:
1. I’m not sure Fig. 7 is needed. It just shows the fluctuations of resistance over time (a clearly understood behavior with FSR technologies).
2. Some equations are numbered (equation 1 and 1.1) while others are not (transfer functions). I suggest consistent formatting.
3. Pneumatic actuators are powerful in general, however, their actuation mechanism makes them bulky and less suitable for haptic interactions (specially for wearable applications). I suggest the Authors add a paragraph summarizing the limitations of the presented design.
4. I wonder if the Authors would publish the 3D printed design online and make it available to the research community.

Author Response

The paper presents the design and characterization of an integrated soft haptic actuator and sensor. The design is 3D printable and is actuated pneumatically. Modeling and characterization demonstrate the ability of the proposed design to provide perceivable force feedback (tactile and proprioceptive). While the design is clearly presented and the characterization is sufficient (force feedback, sensitivity, linearity, SNR, hysteresis, etc.), there are several concerns that must be addressed before publishing this study. Here is a summary of my comments (I lean more towards rejecting the paper than accepting it due to the novelty issue):

The reviewer’s constructive feedback and suggestions to improve the manuscript are highly appreciated.

Major comments:
1. The novelty of the design seems rather incremental. For example, I wonder what the fundamental differences between the proposed design and the one presented in the following study: https://www.mdpi.com/1424-8220/22/10/3779. It seems to have a very similar design (including an overlap of the characterization). This previous study is not even cited in the manuscript!

The authors appreciate for the issues raised by the reviewer and feedback to improve the quality of the manuscript submitted. Our previous pilot study (i.e., https://www.mdpi.com/1424-8220/22/10/3779) focus on influencing/improving postural balance by utilizing multiple soft haptic units. In that study external force sensitive resistors (FSR) were employed to obtain postural data of the subjects. In addition, the previous study only focuses on tactile sensory substitution to improve postural balance. On the contrary, the manuscript submitted focuses on extensive characterization, both mechanically and electrically to assess its performance and a dynamic model is developed to predict its force output for given pressure inputs, and validation of the characterized SoHapS with demonstrations of its capacity to mimick biofeedback in both form of tactile feedback as gripping force and proprioceptive feedback as finger flex-extension position in the context of teleoperation. The article suggested is also included in the revision of the manuscript with further details provided. (Detailed amendments can be referred in the last paragraph on page 3).

  1. Putting the previous study aside, I also wonder what the fundamental contributions to the field of haptics/soft robotics are. While having a 3D printed design definitely brings convenience/saving in the fabrication process, however in terms of actuator/sensing functions, what is novel in the proposed design is not entirely clear to me.

The reviewer’s comments are appreciated, and further clarifications are provided in the introduction to augment clarity of contributions. The soft haptic devices developed aims to provide multi-form of biofeedback while utilizing FDM based additive manufacturing for ease of access, ability to customise for needs from users such as size, stiffness, color, shape so on, lower cost and shorter fabrication cycle, with ability of monolithic fabrication reducing complex assembly issues. In addition of realizing a monolithic soft haptics using pneumatics and self force sensing, two pilot tests employed to communicate two distinct haptic percepts to the user which are tactile feedback and proprioceptive feedback demonstrating adaptability of the soft haptic device to different biofeedback methods. The main contributions can be summarized as i) combining mechanical and electrical characterization, ii) obtaining a dynamic model for the soft haptic device to estimate its force output and iii) validating the model through biofeedback applications for tactile and proprioceptive feedback in the form of soft mechano-tactor. The manuscript amendments are as follow:

“The main contributions of this study are i) the design and development of a unique soft haptic system, SoHapS, which combines pneumatic soft actuation with soft force sensing capabilities to deliver a haptic interface with safe HCI, ii) combining mechanical and electrical characterization, iii) obtaining a dynamic model for the soft haptic device to estimate its force output and iv) validating the static and dynamic characteristics through biofeedback applications for tactile and proprioceptive feedback in the form of soft mechano-tactor. The Perception of pressure applied onto the skin of a subject also depends on the location of the body.” and

“In one of our previous pilot studies, we introduced concept of using soft haptic device to improve postural balance of subjects in which multiple soft haptic units combined with external force sensitive resistors (FSR) [50]. In this study, we demonstrate the efficacy of the SoHapS through not only tactile but also kinesthetic sensory substitution with use of static and dynamic characteristic identified. The major focus is centered on extensive characterization of the soft haptic biofeedback device; both mechanically and electrically to assess its performance, and a dynamic model is developed to predict its force output for given pressure inputs. The characterized soft haptic biofeedback device was then validated with demonstrations of its capacity to mimick biofeedback in the form of tactile feedback as gripping force and proprioceptive feedback as finger flex-extension position in the context of teleoperation.”

  1. Section 4 evaluates the SoHapS device with human subjects, there are several questions/concerns. First of all, how many participants were recruited for the study (only 2?)? If only two subjects were recruited, then Figure 13 would not have statistical significance/ What are the demographics of the test subjects (gender, age group, hand orientation, etc.)? Was the study approved by an appropriate IRB committee? These questions must be addressed before properly analyzing the presented results.

This study focuses on extensive characterization of the pneumatic soft haptic device combining mechanical and electrical properties and obtaining a dynamic model for the soft haptic device to estimate its force output. Validation experiments were conducted to show efficacy of the model in two distinct haptic percepts: i) tactile feedback and ii) proprioceptive feedback demonstrating adaptability of the soft haptic device to different biofeedback methods. Two participants, as couple, were used to conduct biofeedback experiments that one of the subjects acted as master providing an action while the secondary subject received the tailored mimicked biofeedback from SoHapS device. Statistical analyses are aimed to be employed in future studies of the SoHapS with a larger group of people with distinct demographics: gender, age group, hand orientation, etc as suggested by the reviewer. (Detailed amendments can be referred in the last paragraph on page 3).

  1. The Authors claim that the design is capable of rendering tactile and force feedback. In order to render tactile feedback, vibrotactile stimulation should also be possible. To my understanding, that’s not true for the presented design (would it be possible to generate vibration feedback at a frequency of 300 Hz!?).

This study focuses on mimicking tactile feedback with alternative means of external feedback delivered to the subject rather than attempting to replicate tactile feedback in higher frequencies. The SoHapS substitutes biofeedback with use of mechanical pressure intensity rather in contrast to vibro-tactile approach. In this case, amplitude play importance. The reviewer’s concerns are much appreciated on this matter which are also explained in the manuscript with amending the manuscript to eliminate unnecessary misleading leads. (Detailed amendments can be referred in the last paragraph on page 12).

  1. The Authors claim that the device provides a “”high range of force/pressure”feedback. However, the results demonstrate linearity up to around 150 kPa (equivalent to less than 8N). Is this really high force for a kinesthetic haptic device?

Our studies show that bearable pressure, indenting force, applied to the subject is lower than 150kPa. Moreover, bearable force depends on where the SoHapS is strapped on the skin of the subject. further explanation is provided in the manuscript.

  1. One fundamental requirement in the design of force feedback devices is its ability to generate multiple DoF force feedback. The current design provides a single DoF (to my understanding). Any thoughts on the potential of the proposed approach to extend to multiple DoF, including torque feedback?

The authors agree on the point raised by the reviewer. Our future studies will include exploring multiple DoF SoHapS to communicate different biofeedback mimicked. Such approach can augment especially therapy of neurological diseases such as stroke where variety of biofeedback play significant role.

  1. It is not clear how the electric circuit is connected to the SoHapS device (in order to actively actuate it or to read the force measurement). Fig. 12 shows the device but does not show the connection to the control circuitry.

Necessary clarification has been made in the manuscript. The soft force sensor is instrumented with a voltage divider to obtain its resistance values which then correlated to the pressure applied and force generated at the tip of the SoHapS. The soft sensor is connected to the circuit from sensor base and sensor head. As exhibited in Fig.2e-f, contact area affects the resistance values of the soft force sensor.

Minor comments:
1. I’m not sure Fig. 7 is needed. It just shows the fluctuations of resistance over time (a clearly understood behavior with FSR technologies).

Fig.7 shows electrical resistance of the soft force sensor of the SoHapS. The manuscript does not provide electrical resistance of the FSR sensor used in biofeedback experiments for constructing force cube.

  1. Some equations are numbered (equation 1 and 1.1) while others are not (transfer functions). I suggest consistent formatting.

Suggested corrections have been made.

  1. Pneumatic actuators are powerful in general, however, their actuation mechanism makes them bulky and less suitable for haptic interactions (specially for wearable applications). I suggest the Authors add a paragraph summarizing the limitations of the presented design.

The reviewer’s suggestion is much appreciated. Limitations of use of pneumatic actuation, in particular, in soft robotic applications are highlighted in the manuscript.

  1. I wonder if the Authors would publish the 3D printed design online and make it available to the research community.

The authors have considered making the design open source through one of the soft robotics community platforms which can bring further engagement.

Reviewer 2 Report

The article is well detailed and well written, however I have some questions that the authors should address:

1. In figure 2 the legend is missing.

2. Can the author detail the value of the parameters used in Mooney-Rivlin for both materials (TPU and conductive TPU)? Although it is true that another article is referenced, since it is not open access, it makes it difficult to verify and replicate the study.

3. Line 197: How the Contact Pairs have been modelled? Can the authors give some details, such as coefficient of friction used, etc.

4. Figure 3 and 12a, choose another color for the indicator lines and change the color to a more striking color that stands out against the background.

5. Figures 4, 5, 8 and 10 have no label X axis.

6. What is the meaning of the letters w, o, e in Figure 13a?

7. In Figure 13b, how does the author interpret the negative finger flexion values? And how can it be interpreted by the other subject if, as I understand, the actuator works only with positive pressure?

Author Response

The article is well detailed and well written, however I have some questions that the authors should address:

The reviewer’s constructive feedback and suggestions are well appreciated.

  1. In figure 2 the legend is missing.

The authors appreciate for this important issue which aids identifying values in the figure. For the compactness of the figure as high number of sub-elements are exhibited in the figure, maximum values are given in the figure caption.

  1. Can the author detail the value of the parameters used in Mooney-Rivlin for both materials (TPU and conductive TPU)? Although it is true that another article is referenced, since it is not open access, it makes it difficult to verify and replicate the study.

The manuscript has been amended with the details of material models, using Mooney-Rivlin, for both materials; TPU and conductive TPU.

  1. Line 197: How the Contact Pairs have been modelled? Can the authors give some details, such as coefficient of friction used, etc.

Further details have been provided in the manuscript to clarify the issue raised.

  1. Figure 3 and 12a, choose another color for the indicator lines and change the color to a more striking color that stands out against the background.

Figure 3 and 12a have been amended with the reviewer’s suggestion to improve the visibility of the information communicated in the figures.

  1. Figures 4, 5, 8 and 10 have no label X axis.

Figures 4, 5, 8 and 10 have multiple figures thus for compactness of figure presentation, single mutual x-axis label is provided. For consistency, Figure 6 is also amended.

  1. What is the meaning of the letters w, o, e in Figure 13a?

w depicts weighted cube gripping, e depicts empty cube gripping, and o depicts no gripping. This information is provided in the figure caption and highlighted.

  1. In Figure 13b, how does the author interpret the negative finger flexion values? And how can it be interpreted by the other subject if, as I understand, the actuator works only with positive pressure?

The reviewer’s point is valid which leads to confusion. Subject 1, dictates the finger angular position, has initial values starting from -10° and finger flex-extensions up to 90° thus the manuscript has been amended accordingly to clarify the point “… finger angular positions within a range of -10° - 90° where SoHapS bio-mimicked feedback pressure was mapped to the range of finger angular positions in which -10° was communicated as no pressurization.”

Reviewer 3 Report

This study proposed a unique soft monolithic haptic feedback device (SoHapS) that is directly manufactured using a low-cost and open source fused deposition modeling (FDM) 3D printer by employing a combination of soft conductive and nonconductive thermoplastic polyurethane (TPU) materials. This work is complete, and it is one of the most important research areas. I think this paper can be accepted at its current form.

Author Response

This study proposed a unique soft monolithic haptic feedback device (SoHapS) that is directly manufactured using a low-cost and open source fused deposition modeling (FDM) 3D printer by employing a combination of soft conductive and nonconductive thermoplastic polyurethane (TPU) materials. This work is complete, and it is one of the most important research areas. I think this paper can be accepted at its current form.

The authors appreciate for constructive feedback and complementary comments of the reviewer.

Round 2

Reviewer 1 Report

The efforts of the Authors to address the provided comments are greatly appreciated. However, there are a few pending concerns that should be addressed:

1. If the human subject experiment is completed with only two subjects, I suggest removing this part entirely and focus on the mechanical/electrical characterization. There are also other details about the human subject experiment that remain unaddressed (IRB approval?).

2. The Authors should be clear about the inability of the proposed device to induced vibration feedback. I therefore suggest revising the language to clarify this point - tactile feedback includes vibration. 

Author Response

The efforts of the Authors to address the provided comments are greatly appreciated. However, there are a few pending concerns that should be addressed:

The authors appreciate for the reviewer’s continuing efforts and time to improve the quality of this study.

  1. If the human subject experiment is completed with only two subjects, I suggest removing this part entirely and focus on the mechanical/electrical characterization. There are also other details about the human subject experiment that remain unaddressed (IRB approval?).

The authors agree with the reviewer on the point of statistical analyses of the device proposed for human trials of participants from diverse demography. The pre-trial tests have been employed to demonstrate the purpose of the proposed device in order to validate the main contributions of this study by the use of characterized static and dynamic properties. In addition, the role of one of the two subjects is to create signal/data to communicate to the actual subject wearing the device, while efficacy of the method is assessed from action of the actual subject (S2). As outlined in the manuscript, next stage is to test the device with number of participants with diverse demography which is also suggested by the reviewer with IRB approval. An IRB approval have been obtained to be utilized for next stage tests with larger number of participants. Further, the title may be creating a confusion about “inducing” tactile feedback and/or proprioceptive feedback which has been amended to eliminate this confusion. The soft haptic device substitutes the tactile and proprioceptive sensation with mechanical pressure applied onto the skin of the wearer.

            As highlighted in the manuscript, this study focus on mechanical/electrical characterization and validation of the characterization with two different pre-trial tests. “The main contributions of this study are i) the design and development of a unique soft haptic system, SoHapS, which combines pneumatic soft actuation with soft force sensing capabilities to deliver a haptic interface with safe HCI, ii) combining mechanical and electrical characterization, iii) obtaining a dynamic model for the soft haptic device to estimate its force output and iv) validating the static and dynamic characteristics through biofeedback applications for tactile and proprioceptive feedback in the form of soft mechano-tactor in the form of mechanical pressure rather than vibration.” The manuscript has further been amended in the validation section, Section 4, to highlight the major contributions and the use of pre-trials for validation of the static and dynamic characterization as “We have developed two different demonstration tasks for validating the static and dynamic characteristics through biofeedback applications for tactile and proprioceptive feedback in the form of soft mechano-tactor in the form of mechanical pressure and to highlight the efficacy of the SoHapS for inducing substitute bio-mimicked feedback.” Section 4 title has also been amended to be in line with the amendments and purpose of the pre-trial tests.

  1. The Authors should be clear about the inability of the proposed device to induced vibration feedback. I therefore suggest revising the language to clarify this point - tactile feedback includes vibration.

The reviewer’s constructive feedback is much appreciated. The manuscript has been amended accordingly to clarify the point that the device proposed creates mechanical pressure applied onto the skin of the subject rather it does not create vibration. Thus, the manuscript has been amended against the use of “inducing” to “substituting” the tactile and proprioceptive sensation. The title has also been amended to eliminate the confusion of creating a tactile sensation.